# SM-ShapNAS: Shapley-Enhanced Multimodal Neural Architecture Search via Sparse Modeling

## Abstract

Despite their excellent performance on various multimodal learning tasks, deep neural networks (DNNs) are often characterized as "black boxes". Some techniques aid in designing explainable DNNs. For instance, sparse modeling limits the sparsity while preserving key features, and the Shapley value from game theory quantifies the true contribution of each component, both of which are recognized for their strong explainability. However, designing explainable multimodal DNNs by manually designing unimodal backbones and multimodal feature fusion models requires substantial expertise and time. This paper proposes a novel multimodal neural architecture search (NAS) method, termed Shapley-Enhanced Multimodal Neural Architecture Search via Sparse Modeling (SM-ShapNAS), for automating the design of appropriate and explainable multimodal DNNs. SM-ShapNAS incorporates sparse attention and sparse convolutional operations within a predefined search space, and uses the Shapley value approximated by group policy to evaluate the true contribution of each operation in the fusion cells. By combining sparse modeling and the Shapley value, the proposed SM-ShapNAS automatically generates efficient and explainable multimodal DNNs. Experimental results on three multimodal datasets demonstrate that the SM-ShapNAS achieves competitive performance compared to the state-of-the-art multimodal NAS methods, particularly in noisy environments.

## 1 Introduction

With the rapid development of multimodal learning, designing efficient networks to integrate heterogeneous data modalities has become a research focus, such as action recognition (Bruce et al., 2022) and cross-modal retrieval (Hu et al., 2021; Wang et al., 2024). While deep neural networks (DNNs) has made remarkable success in designing high-performance models, it has been increasingly realized and criticized that many DNNs lack theoretical support. Different from "interpretability" which refers to a model to be understood due to its inherent simplicity or transparent structure, we focus on "explainability", which emphasizes techniques to generate comprehensible rationales for "black box" models. For example, the goal of layer operations such as convolution, pooling, and normalization is to minimize the training loss, which results in unreasonable middle layers for DNNs. The lack of explainability hinders enhancing learning systems for noisy data (Xua & Yang, 2024). Most existing multimodal DNNs only focus on model performance, which makes it difficult to discover internal relationships between different modalities and make reasonable decisions.

However, manually designing explainable and efficient architectures for multimodal tasks remains challenging, as it requires significant expertise to balance modality-specific processing, cross-modal fusion, and computational efficiency. This limitation has stimulated the rise of neural architecture search (NAS), which has emerged to automate the design of efficient neural networks. NAS is an automatic method for searching the optimal neural architecture within a predefined search space (Bello et al., 2017). Some works apply NAS to multimodal learning, for example, MFAS (Pérez-Rúa et al., 2019) proposes a sequence model-based adaptive search method, with the challenge that the single fusion operation leads to a limited combination for feature fusion strategies when dealing with multiple modalities. BM-NAS (Yin et al., 2022) adopts an efficient bilevel multimodal architecture search scheme. However, the magnitude of architecture parameters fails to reflect the

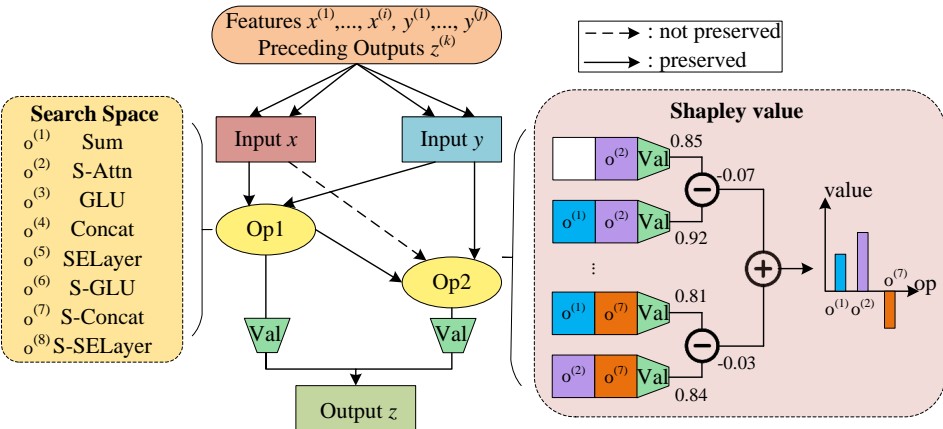

Figure 1: A feature fusion cell in SM-ShapNAS. Two inputs are selected from unimodal features and preceding outputs. The inner-cell operations are chosen from normal operations and sparse operations in the search space. Then we employ the Shapley value via group policy to evaluate the true contribution of operations according to the validation accuracy difference.

true contribution of each operation (Wang et al., 2021; Zhou et al., 2021), thereby hindering the performance of derived architectures. DC-NAS (Liang et al., 2024) exchanges knowledge from two small knowledge bases, but it may lead to information loss on large datasets.

To address these challenges, this paper proposes a novel explainable multimodal NAS framework that integrates sparse modeling and the grouped Shapley value. We design sparse attention and sparse convolution operations in the predefined search space, to improve the feature fusion capability and improve search efficiency. A cell in the search process in SM-ShapNAS is shown in Fig. 1. Each cell has two inputs and one output. They choose intra-cell and inner-cell operations to discover proper architectures by evaluating the operation contribution. In multimodal NAS, the function of a single operation is often non-independent, and its efficiency is highly dependent on the dynamic coupling and context collaboration between modalities. For example, the feature extraction of visual modality may adjust the weight distribution due to semantic constraints of text modality, while the design of the cross-modal fusion layer needs to be optimized synchronously to match different modalities. We apply grouped Shapley value instead of the magnitude of the gradient descent architecture parameter when evaluating the contribution of operations to the architectures. Furthermore, since directly calculating Shapley value is an NP-hard problem, we employ group policy to estimate the Shapley value, enabling the evaluation of the true contribution of each operation.

The main contributions of this paper are four-fold.

1. We design a search space, including normal and sparse operations (sparse attention and sparse convolution operations), which improves the feature extraction capability.

2. To efficiently determine the true contribution of each operation, we estimate the Shapley value using group policy to evaluate candidate operations.

3. We propose a novel multimodal NAS framework, termed SM-ShapNAS, to automatically design explainable multimodal DNNs.

4. Experiments on three multimodal benchmarks demonstrate that SM-ShapNAS outperforms state-of-the-art peer methods, particularly in noisy environments.

## 2 RELATED WORKS

### 2.1 EXPLAINABLE MULTIMODAL NAS

Recently, several multimodal NAS methods have attracted significant attention due to their ability to automatically identify the optimal architectures. MFAS (Pérez-Rúa et al., 2019) treats multimodal

fusion as a NAS problem, it proposes a new search space and employs sequential model-based optimization (SMBO) algorithm. BM-NAS (Yin et al., 2022) introduces a bilevel scheme to search for both the unimodal feature selection strategy and the fusion strategy. However, its narrow search space leads to suboptimal results. DC-NAS (Liang et al., 2024) utilizes the small knowledge bases to improve search efficiency. CoMO-NAS (Fu et al., 2024) applys core structures to guide the Pareto Frontier search. These methods all suffer from the problem of lack of explainability, which makes it difficult for researchers to understand their working principle and decision-making basis.

Some strategies have attracted researchers' attention due to their strong theoretical explainability. Sparse modeling has emerged as an important technique for enhancing the explainability of networks (Li et al., 2022). In multimodal learning, sparse modeling has been widely applied in various tasks, such as image fusion (Veshki & Vorobyov, 2022), finger recognition (Li et al., 2021) and so on (Wu et al., 2013; Scetbon et al., 2021; Lecouat et al., 2020). CCFL (Veshki & Vorobyov, 2022) uses separate convolutional sparse coding to approximate shared and indenpendent features. SDMFC (Li et al., 2021) constructs an overcomplete dictionary on which the extracted multimodal features are sparsely encoded. Recently, the DeepSeek-AI team (Yuan et al., 2025) proposed the native sparse attention, has greatly increased research interest. Notably, in noise imaging scenarios, sparse design reduce interference from corrupted regions while providing explainability through their sparsity.

## 2.2 SHAPLEY VALUE

The Shapley value (Shapley et al., 1953; Ghorbani & Zou, 2019) which rooted in cooperative game theory, have been adapted to explain model predictions by attributing contributions to input features. (Lundberg & Lee, 2017) proposes a unified framework SHAP, which unifies several feature attribution methods under a theoretical framework and uses the Shapley value to assign the contribution of each feature to the model prediction. It ensures the consistency and local accuracy of the interpretation, and provides a theoretical basis for model interpretation. (Shanbhag et al., 2021) introduces a model-agnostic framework to quantify the contribution of input features to the prediction drift. Shapley-NAS (Xiao et al., 2022) first employs the Shapley value into unimodal NAS, however, calculating the Shapley value for only normal operations is difficult to effectively reduce the computational cost and does not perform well enough in noisy environment.

This paper designs an efficient and explainable multimodal NAS method that uses sparse modeling in the predefined search space and grouped Shapley value to evaluate the true performance of each operation. The proposed SM-ShapNAS follows the alternating optimization paradigm in differentiable architecture search (Liu et al., 2019).

## 3 METHOD

The overall framework of the proposed SM-ShapNAS is shown in Fig. 2. The features are extracted using predefined unimodal backbone. Then we search for feature fusion cells in the search space including normal operations and sparse operations, and employ grouped Shapley value to evaluate the contribution. Each feature fusion cell represented by a directed acyclic graph (DAG) has two inputs that are selected from the unimodal features and the preceding outputs. We estimate the Shapley value to identify true contribution of each operation by group policy, which determines the contribution based on the validation accuracy change.

Algorithm 1 shows the search process, which follows DARTS to alternatively optimize architecture parameters and model weights, and update the architecture according to grouped Shapley value. The proposed multimodal NAS method involves two core concepts: sparse coding and grouped Shapley value, which are closely integrated.

### 3.1 SPARSE MODELING

To address noise in sequential data and improve model explainability, we introduce a 1-dimensional sparse convolution that extends the principles of sparse feature learning to 1-dimensional sequence. For input vector $\boldsymbol{x} = (x_1, x_2, \ldots, x_c) \in \mathbb{R}^{C \times L}$, and the convolution dictionary $\boldsymbol{A} \in \mathbb{R}^{C \times C \times k}$, where $C$ is the number of channels, $L$ is the sequence length, $k$ is the 1-dimensional kernel length. Then sparse output vector can be expressed as $\boldsymbol{z} \in \mathbb{R}^{C \times L}$.

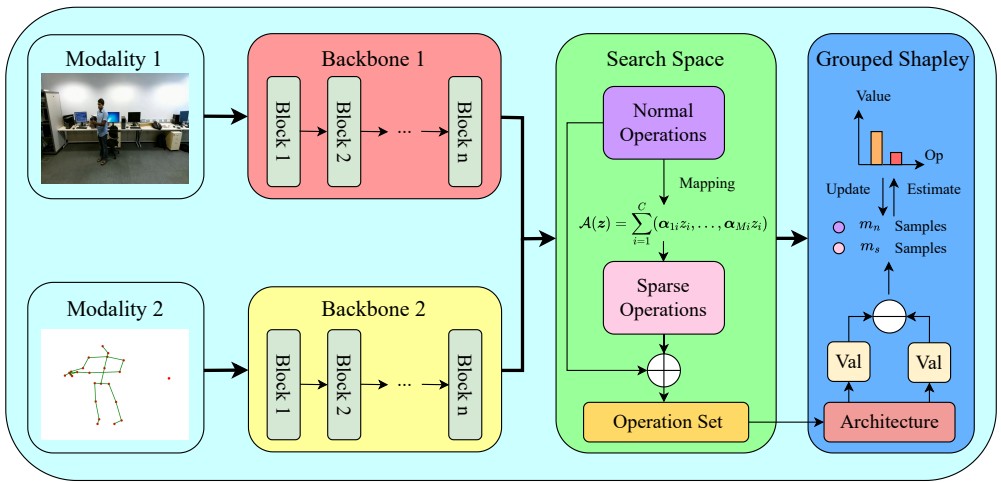

Figure 2: Overall framework of the proposed SM-ShapNAS.

---

**Algorithm 1** Pseudocode of the proposed SM-ShapNAS.

---

**Require:** The search space $\mathcal{O}$.
**Ensure:** The optimal fusion architecture *architecture_best*.
1: Initialize architecture parameters $\alpha$ and model weights $\omega$
2: Initialize *architecture_best* based on $\alpha$ and $\omega$ in the predefined search space $\mathcal{O}$
3: Evaluate individuals in $P$ by training architectures encoded by the individuals for $M$ epochs and compute the fitness;
4: **while** not converged **do**
5:    $\omega \leftarrow$ Update $\omega$ by optimizing $\mathcal{L}_{\text{train}}$
6:    $\alpha \leftarrow$ Update $\alpha$ by optimizing $\mathcal{L}_{\text{val}}$
7:    $\phi \leftarrow$ Calculate current grouped Shapley value $\phi$
8:    Construct *architecture* based on $\phi$ and $\omega$ in the predefined search space $\mathcal{O}$
9:    *architecture_best* $\leftarrow$ Update *architecture_best* using the *architecture*
10: **end while**
11: return $P$;

---

As a sparse coding layer, it is used to perform an inverse mapping to a preferably sparse output vector $z = (z_1, z_2, \ldots, z_c)$. The convolution dictionary $A \in \mathbb{R}^{C \times C \times L}$ can be expressed in the following form:

$$A = \begin{pmatrix} \boldsymbol{\alpha}_{11} & \boldsymbol{\alpha}_{12} & \boldsymbol{\alpha}_{13} & \ldots & \boldsymbol{\alpha}_{1C} \\ \boldsymbol{\alpha}_{21} & \boldsymbol{\alpha}_{22} & \boldsymbol{\alpha}_{23} & \ldots & \boldsymbol{\alpha}_{2C} \\ \vdots & \vdots & \vdots & \ddots & \vdots \\ \boldsymbol{\alpha}_{C1} & \boldsymbol{\alpha}_{C2} & \boldsymbol{\alpha}_{C3} & \ldots & \boldsymbol{\alpha}_{CC} \end{pmatrix} \in \mathbb{R}^{C \times C \times k},$$

where $\boldsymbol{\alpha}$ represents 1-dimensional kernel of length $k$. Then input vector $\boldsymbol{x}$ is

$$\boldsymbol{x} = \mathcal{A}(\boldsymbol{z}) = \sum_{i=1}^{C} (\boldsymbol{\alpha}_{1i} z_i, \ldots, \boldsymbol{\alpha}_{Mi} z_i) \in \mathbb{R}^{C \times L} \tag{1}$$

To accelerate convergence and achieve end-to-end sparsity, we use fast iterative shrinkage-thresholding algorithm (FISTA) for forward propagation. For the sparse convolutional layer, we learn the sparse coding to minimize the following loss:

$$\mathcal{L} = \frac{1}{2} \|\boldsymbol{x} - \mathcal{A}(\boldsymbol{z})\|_2^2 + \lambda_1 \|\boldsymbol{z}\|_1 + \lambda_2 \|\boldsymbol{z}\|_2^2 \tag{2}$$

where $\lambda_1$ and $\lambda_2$ are regularity coefficients, $\mathcal{A}$ is the convolutional dictionary. $\lambda_1 \| \;\|_1$ and $\lambda_2 \| \;\|_2^2$ are $\ell_1$ and $\ell_2$ regularization terms respectively.

## 3.2 SEARCH SPACE

**Inter-cell.** By adopting the continuous relaxation mechanism in DARTS, all predecessors of cells are dynamically connected through a fully directed supernet. For any two nodes $n^{(i)}, n^{(j)} (i < j)$, the output of edge $(i, j)$ is defined as:

$$x^{(j)} = \sum_{i \to j} \sum_{f \in \mathcal{F}} \frac{\exp\left(\alpha_f^{(i,j)}\right)}{\sum_{f' \in \mathcal{F}} \exp\left(\alpha_{f'}^{(i,j)}\right)} \cdot f\left(x^{(i)}\right) \tag{3}$$

where $\alpha_f^{(i,j)}$ is the architecture parameter of edge $(i, j)$, operation $f$ is selected from function set $\mathcal{F}$. Specifically, $f(x) = x$ preserves the edge, whereas $f(x) = 0$ removes the edge.

In the search process, the architecture parameters $\alpha$ of the candidate operations are alternately optimized through gradient descent in conjunction with the network weights $w$, with the goal of minimizing validation loss:

$$\min_{\alpha} \mathcal{L}_{\text{val}}\left(w^*(\alpha), \alpha\right), \quad \text{s.t.} \quad w^*(\alpha) = \arg\min_{w} \mathcal{L}_{\text{train}}\left(w, \alpha\right) \tag{4}$$

In the evaluation process, the final architecture is determined through discretization. The operation with the highest weight on edge $(i, j)$ is retained:

$$f^* = \arg\max_{f \in \mathcal{F}} \alpha_f^{(i,j)} \tag{5}$$

**Intra-cell.** As the scheme in DARTS, Each cell represents a DAG that comprises a set of inner nodes, namely two input nodes, one output node, and several middle nodes. Operations of these nodes are selected from the predefined search space in Table 1. We set $x, y$ as two inputs, and $z$ as one output. The different operations are described in detail in *Appendix A*.

Table 1: The proposed search space contains normal operations and sparse operations.

| Type | Unimodal Backbone | | | |
|------|-------------------|---|---|---|
| Normal | Sum | Gated Linear Units | Concat | SELayer |
| Sparse | Sparse-Attention | Sparse-GLU | Sparse-Concat | Sparse-SELayer |

Similar to the inter-cell continuous relaxation mechanism, the output $z^{(t)}$ at node $t$ can be represented as below.

$$z^{(t)} = \sum_{o \in \mathcal{O}} \frac{\exp\left(\alpha_o^{(t)}\right)}{\sum_{o' \in \mathcal{O}} \exp\left(\alpha_{o'}^{(t)}\right)} \cdot o\left(x^{(i)}, y^{(j)}\right) \tag{6}$$

where $x^{(i)}$ and $y^{(j)}$ are the input $x$ at preceding node $i$, the input $y$ at preceding node $j$ respectively, operation $o$ is selected from the predefined search space $\mathcal{O}$.

## 3.3 GROUPED SHAPLEY VALUE

Differentiable NAS couples architecture parameter $\alpha$ with weights $\omega$ during the joint optimization process, and $\alpha$ fails to reflect true contribution of the operations. We address this by applying grouped Shapley value, a solution based on fair distribution from cooperative game theory, to evaluate the component contribution. We map NAS as a cooperative game where players are candidate inter-cell and intra-cell operations, coalition forms architecture, payoff function corresponds to validation performance, and payoff allocation quantifies contribution. In the cooperative game, players set is $N = \{1, 2, \ldots, n\}, n = \text{Ops}_{\text{inter}} \cdot \text{Ops}_{\text{intra}}$, feature value function $v : 2^N \to \mathbb{R}$ represents performance metric for each any subset of players $\mathcal{S} \subseteq N$, and the goal is to find the payoff allocation vector $\phi \in \mathbb{R}^n$:

$$\sum_{i=1}^{n} \phi_i = v(N) \tag{7}$$

The Shapley value is the fairest payoff allocation scheme in the cooperative game, which calculates the expected value of the true contribution of operation $o_i$:

$$\phi_i = \sum_{k_g=1}^{K_g} \frac{P(group = k_g)}{m_k} \sum_{\mathcal{S} \subseteq N \setminus \{o_i\}} \frac{|\mathcal{S}|!(n - |\mathcal{S}| - 1)!}{n!} [v(\mathcal{S} \cup \{o_i\}) - v(\mathcal{S})] \tag{8}$$

where $\mathcal{S}$ is the set of predecessors of operation $o$ in a given permutation. $k_g$ is the operation group. $m_k$ is the initial samples for group $k_g$. In this paper, we utilize the change in validation accuracy as the feature value function to evaluate the operations. Considering that the feature value function $v(\cdot)$ needs to be evaluated exactly for each subnetwork, and the computational complexity $O(2^N)$ is too high, the Shapley value is approximated by means of group policy and Monte Carlo sampling. This way, the calculation complexity reduces from $O(N \cdot 2^N)$ to $O(mN)$, where $m$ is the number of samples. Then the estimated Shapley value can be expressed as below.

$$\phi_i \doteq \mathbb{E}_{\pi \sim \Pi} \left[ v\left(\mathcal{S}_{\pi,i} \cup \{o_i\}\right) - v\left(\mathcal{S}_{\pi,i}\right) \right] \tag{9}$$

where $\Pi$ is the set of all operation permutations, $\mathcal{S}_{\pi,i}$ is the set of all operations before operation $o_i$ in permutation $\pi$.

The grouped Shapley value in NAS is to represent the contribution of each operation, and the search target is modified to:

$$\alpha \propto \phi\left(\mathcal{L}_{\text{val}}\left(w^*, \alpha\right)\right) \quad \text{s.t.} \quad w^* = \arg\min_w \mathcal{L}_{\text{train}}(w, \alpha) \tag{10}$$

where $\alpha$ is architecture parameter, $w$ is network weight, $\phi$ is grouped Shapley value. Then we update $\alpha$ based on Shapley value estimated by group policy and Monte-Carlo sampling.

$$\alpha_t = \alpha_{t-1} + \epsilon \cdot \frac{s_t}{\|s_t\|_2} \tag{11}$$

$$s_t = \frac{\phi\left(\text{Acc}_{\text{val}}\left(w_{t-1}, \alpha_{t-1}\right)\right)}{\|\phi\left(\text{Acc}_{\text{val}}\left(w_{t-1}, \alpha_{t-1}\right)\right)\|_2} \tag{12}$$

where $\alpha_t$ is the architecture parameter of step $t$, $\epsilon$ is the step size, $s_t$ is grouped Shapley value at step $t$, and $\|\|_2$ is $\ell_2$ norm. Note that $m_k$ is only for initialization, and the number of samples changes proportionally according to the architecture parameters for each operation after warmup.

## 4 EXPERIMENTS

In this section, we conduct extensive experiments to evaluate the proposed SM-ShapNAS across three multimodal tasks. We use the multilabel movie genre classification dataset MM-IMDB (Arevalo et al., 2017), the multimodal action recognition dataset NTU RGB+D (Shahroudy et al., 2016), and the multimodal gesture recognition dataset EgoGesture (Zhang et al., 2018).

### 4.1 DATASETS AND TRAINING SETTINGS

MM-IMDB is a multilabel classification dataset containing 25,959 movies. We adopt VGG Transfer (Simonyan & Zisserman, 2014) and Maxout MLP (Goodfellow et al., 2013) as the backbones for image modality and text modality, respectively. NTU RGB+D is a large-scale multimodal dataset for human action recognition. It contains 56,880 motion samples from 40 subjects covering 60 classes of movements. We use Inflated ResNet-50 (Baradel et al., 2018) for the video modality, and Co-occurrence (Chen et al., 2020) for the skeleton modality. EgoGesture contains 2,081 RGB-D videos, 24,161 gesture samples, and 2,953,224 frames collected from 50 distinct subjects across 83 classes for gesture recognition. ResNeXt-101 (Köpüklü et al., 2019) serves as the backbones for both RGB and depth video modalities.

We set the regularity coefficients $\lambda_1 = 0.1$, $\lambda_2 = 0$ in sparse operations. We use the Adam (Kingma & Ba, 2014) optimizer, learning rate 3e-4, and $\ell_2$ weight decay 1e-4 for architecture parameter optimization. We use the Adam optimizer with Cosine Annealing scheduler, maximum learning rate 1e-3, minimum learning rate 1e-6, and $\ell_2$ weight decay 1e-4 for network parameters. For estimating

grouped Shapley value, initial normal samples $m_n = 20$, initial sparse samples $m_s = 60$, and step size is $\epsilon = 0.1$. The experimental details are placed in *Appendix B*.

Notably, the proposed method focuses on the feature fusion process and relies on unimodal backbones. It prioritizes computational efficiency and task-specific feature extraction but inherently restricts the layer visualization. In particular, the proposed method only requires 0.08 GPU days and 0.80 GPU days respectively on MM-IMDB and EgoGesture to search for the optimal architecture.

## 4.2 COMPARISON WITH STATE-OF-THE-ART METHODS

Table 2, Table 3 and Table 4 show the performance of the proposed SM-ShapNAS on the three datasets respectively. The proposed method adopts the average of 10 results.

Table 2: Multilabel genre classification results on MM-IMDB dataset. F1-Weighted (F1-W) and F1-Macro (F1-M) are reported.

| Method | Modality | F1-W(%) | F1-M(%) |
|---|---|---|---|
| Unimodal Backbone | | | |
| VGG Transfer (ICLR15) | Image | 49.21 | 33.50 |
| Maxout MLP (ICML13) | Text | 57.54 | 45.98 |
| Multimodal Methods | | | |
| MFAS (CVPR19) | Image + Text | 62.50 | 55.68 |
| BM-NAS (AAAI22) | Image + Text | 62.40 | 54.34 |
| DC-NAS (AAAI24) | Image + Text | 63.70 | - |
| CoMO-NAS (MM24) | Image + Text | 63.84 | - |
| SM-ShapNAS (Ours) | Image + Text | **65.54 ± 0.10** | **59.37 ± 0.16** |

In MM-IMDB, we compare SM-ShapNAS with some state-of-the-art peer methods. F1-Weighted calculates the F1-score for each class separately and then takes a weighted average based on true instances counts. It is suitable for imbalanced datasets. F1-Macro calculates the F1-score for each class separately and then averages them equally. It is used when evaluating performance across all classes without bias toward dominant classes. As shown in Table 2, the proposed method has competitive performance in both the F1-W and F1-M metrics.

Table 3: Multimodal action recognition accuracy on NTU RGB+D dataset.

| Method | Modality | Accuracy(%) |
|---|---|---|
| Unimodal Backbone | | |
| Inflated ResNet-50 (CVPR18) | Video | 83.91 |
| Co-occurrence (IJCAI18) | Pose | 85.24 |
| Multimodal Methods | | |
| MFAS (CVPR19) | Video + Pose | 89.50 |
| BM-NAS (AAAI22) | Video + Pose | 90.48 |
| DC-NAS (AAAI24) | Video + Pose | 90.85 |
| CoMO-NAS (MM24) | Video + Pose | 90.94 |
| SM-ShapNAS (Ours) | Video + Pose | **92.76 ± 0.04** |

In Table 3 and Table 4, the optimal architecture searched by the proposed method achieves an accuracy of 91.35% on NTU RGB+D and 95.49% on EgoGesture, which demonstrates the effectiveness of the proposed SM-ShapNAS. We further analyze all the optimal architectures on three datasets in the *Appendix C*.

## 4.3 DISCUSSION

Moreover, the proposed SM-ShapNAS performs well under varying severity levels of noise. Table 5 shows the results under a Gaussian noise environment on three multimodal datasets. We use only the F1-W score as the metric for MM-IMDB because of the imbalanced classes. By simulating physical

Table 4: Multimodal gesture recognition accuracy on EgoGesture dataset.

| Method | Modality | Accuracy(%) |
|---|---|---|
| Unimodal Backbone | | |
| ResNeXt-101 (FG19) | RGB | 93.75 |
| ResNeXt-101 (FG19) | Depth | 94.03 |
| Multimodal Methods | | |
| BM-NAS (AAAI22) | RGB + Depth | 94.96 |
| DC-NAS (AAAI24) | RGB + Depth | 95.22 |
| CoMO-NAS (MM24) | RGB + Depth | 95.25 |
| SM-ShapNAS (Ours) | RGB + Depth | **96.46 $\pm$ 0.05** |

Table 5: Results under different Gaussian noise levels on three multimodal datasets.

| | **MM-IMDB(%)** | | | **NTU RGB+D(%)** | | | **EgoGesture(%)** | | |
|---|---|---|---|---|---|---|---|---|---|
| Noise level [*] | level 1 | level 2 | level 3 | level 1 | level 2 | level 3 | level 1 | level 2 | level 3 |
| BM-NAS | 60.11 | 47.81 | 28.16 | 88.33 | 85.38 | 84.31 | 93.92 | 89.30 | 82.77 |
| SM-ShapNAS | 64.82 | 60.67 | 57.01 | 92.20 | 90.59 | 89.55 | 96.27 | 94.72 | 93.94 |

[*] Level 1, level 2, level 3 correspond to variances of $\sigma_1^2 = 0.01$, $\sigma_2^2 = 0.05$, $\sigma_3^2 = 0.1$, respectively.

perturbation in real scenes, image noise enhancement can directly improve the robustness of the model. Therefore, we only noise the image modalities. As we can see, the proposed method can deal with different severities of Gaussian noise well. We employ pretrained backbones for NTU RGB+D and EgoGesture, while use only the backbone structure for MM-IMDB, which results in a significant performance drop on MM-IMDB, whereas the performance on NTU RGB+D and EgoGesture are relatively flat.

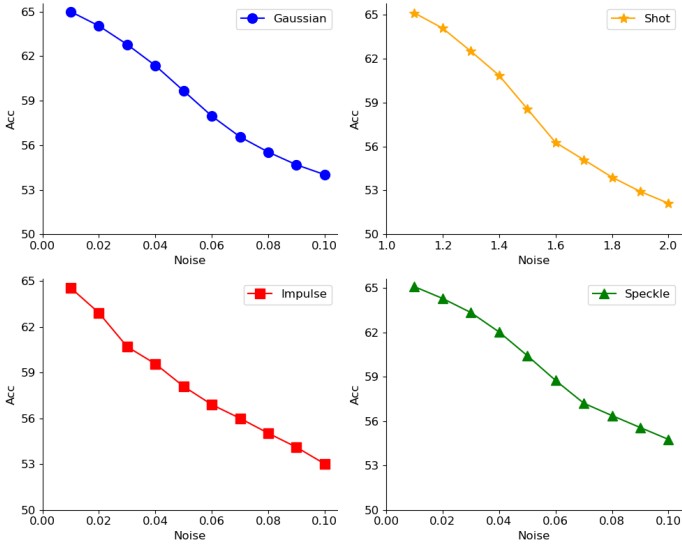

Figure 3: The performance of SM-ShapNAS on MM-IMDB under different additive noises.

Since additive noise is directly superimposed on the data without altering its original sparsity, we report the performance of SM-ShapNAS under different additive noises. As Fig. 3 shown, SM-ShapNAS has good performance under Gaussian, Shot and Impulse noise. We also report the result under Speckle noise to explore the effect of multiplicative noise.

Fig. 4 shows the architecture parameter $\alpha$ of 4 operations that contribute the most in the first fusion cell in MM-IMDB. The Sparse-Attn operation is always one of the top two operations. Although the Sparse-GLU operation performs best in the early stages, the Sum operation exhibits superior architectural utility in terms of architecture parameters at epoch 5 and remains so until the end. Fur-

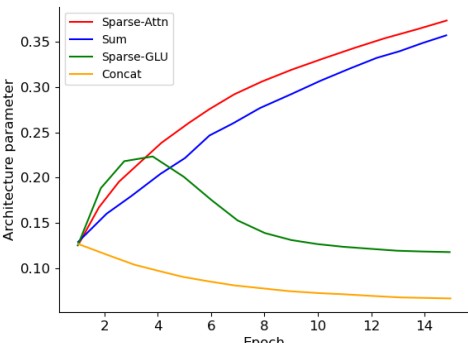

Figure 4: The architecture parameter $\alpha$ in the first fusion cell in MM-IMDB.

thermore, we analyze the results with different backbones and the utilization of different modalities in the *Appendix C*.

## 4.4 ABLATION STUDY

We analyze the effectiveness of sparse modeling and grouped Shapley value are shown in Table 6.

Table 6: Ablation study of different search spaces and grouped Shapley value.

| Method | Search Space | Evaluation | MM-IMDB(%) | NTU(%) | Ego(%) |
|--------|-------------|------------|------------|--------|--------|
| Baseline | Normal | Arch params | 62.42 | 90.38 | 94.90 |
| Shapley | Normal | Grouped Shapley | 64.06 | 91.14 | 95.68 |
| Sparse | Normal + Sparse | Arch params | 63.12 | 91.00 | 94.90 |
| SM-ShapNAS | Normal + Sparse | Grouped Shapley | 65.54 | 92.76 | 96.46 |

In Table 6, the search space containing normal and sparse operations performs better than only normal operations. The result on EgoGesture is not obvious, because there exist some similar features in RGB and Depth modalities. We also compare grouped Shapley value strategy with gradient descent updating architecture parameters, and the results show the effectiveness of grouped Shapley value. Note that the sparse operations can greatly improve model performance on MM-IMDB, grouped Shapley value contributes more to the performance on EgoGesture, because more complex fusion process and higher model performance make it more important to evaluate true contributions.

## 5 CONCLUSION

This paper proposes an explainable multimodal NAS method that integrates sparse modeling and grouped Shapley value from game theory to search explainable neural architectures automatically. To the best of our knowledge, SM-ShapNAS fills the gap in explainable multimodal NAS. Specifically, we design sparse attention and sparse convolution operations to improve the feature extraction capability. To efficiently evaluate the potential architectures, we adapt group policy to estimate grouped Shapley value to evaluate the contribution of each operation, thus allowing architecture parameters to be directly updated with true contribution. We experimentally demonstrate the effectiveness of SM-ShapNAS on three multimodal datasets. The optimal architectures searched by SM-ShapNAS provide competitive results on these datasets.

## 6 ETHICS STATEMENT

We acknowledge that all authors of this work have read and commit to adhering to the ICLR Code of Ethics.

## 7 REPRODUCIBILITY STATEMENT

1. All datasets drawn from this paper are publicly available.

2. Partial code for conducting and analyzing the experiments is submitted as *supplementary material*.

3. All source code required for conducting and analyzing the experiments will be made publicly available upon publication of the paper with a license that allows free usage for research purposes.

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

## A  INTRA-CELL OPERATIONS

The normal operations and sparse operations are described below.

1) Sum:
$$z = x + y \tag{13}$$
Sum represents element-wise addition of input features, which is the same way features are fused in DARTS.

2) Sparse-Attention:
$$z = \mathrm{Softmax}\,(S) \cdot \mathcal{A}_v(y) \tag{14}$$
$$S = \frac{\mathcal{A}_q(x) \cdot \mathcal{A}_k(y)^\top}{\sqrt{d_k}} \tag{15}$$
where $S$ denotes the attention score function, $x$ is regarded as Query, $y$ as Key and Value, $d_k$ is the dimension of $x$ and $y$, $L$ is the sequence length. $\mathcal{A}_q, \mathcal{A}_k, \mathcal{A}_v$ represents the sparse weight matrices of Query, Key and Value, respectively. The Sparse-Attention operation can further optimize the computational complexity of attention operation and enhance the ability to focus on highly correlated regions.

3) Gated Linear Units(GLU):
$$z = (W_1 x + b_1) \odot \sigma(W_2 y + b_2) \tag{16}$$
where $W_1$ and $W_2$ denote the weight matrices of $x$ and $y$, respectively, while $b_1$ and $b_2$ represent the bias terms of $x$ and $y$, respectively. $\sigma$ means Sigmoid function, and $\odot$ is hadamard product.

4) Concat:
$$z = \mathrm{ReLU}(W \cdot \mathrm{Concat}(x, y) + b) \tag{17}$$
where the weight matrix $W$ is used for dimensionality reduction, and b is the bias term, which is also the same way features are fused in MFAS.

5) SELayer:
$$z = W \cdot ([\mathrm{Concat}(x, y)] \odot \sigma$$
$$(W_2 \cdot \mathrm{ReLU}\,(W_1 \cdot \mathrm{AvgPool}\,(\mathrm{Concat}(x, y)) + b_1) + b_2)) + b \tag{18}$$
SELayer explicitly models dependencies between channels via squeeze, excitation and scale operation.

6) Sparse-GLU:
$$z = (\mathcal{A}_1(x) + b_1) \odot \sigma(\mathcal{A}_2(y) + b_2) \tag{19}$$

7) Sparse-Concat:
$$z = \mathrm{ReLU}(\mathcal{A}(\mathrm{Concat}(x, y)) + b) \tag{20}$$

8) Sparse-SELayer:
$$z = \mathcal{A} \cdot ([\mathrm{Concat}(\mathrm{x}, \mathrm{y})] \odot \sigma$$
$$(\mathcal{A}_2 \cdot \mathrm{ReLU}\,(\mathcal{A}_1 \cdot \mathrm{AvgPool}\,(\mathrm{Concat}(\mathrm{x}, \mathrm{y})) + b_1) + b_2)) + b \tag{21}$$
The Sparse-GLU, Sparse-Concat and Sparse-SELayer operations incorporate sparse convolution in GLU, Concat, and SELayer operations.

## B  EXPERIMENT CONFIGURATIONS

In MM-IMDB, the search epochs is set to 15, the training epochs is set to 30, batch size is 32, and dropout of 0.2. We adopt 2 fusion cells and 1 step per cell, channel is 192 and sequence length is 16. In NTU RGB+D, the search epochs is set to 30, the training epochs is set to 50, batch size is 16, and dropout of 0.2. We split the 40 subjects, 1, 4, 8, 13, 15, 17, 19 for training, 2, 5, 9, 14 for validation, other subjects for testing. We adopt 2 fusion cells and 2 steps per cell, channel is 128 and sequence length is 8. In EgoGesture, the search epochs is set to 15, the training epochs is set to 30, batch size is 24, and dropout of 0.2. We adopt 2 fusion cells and 3 steps per cell, channel is 128 and sequence length is 8. The warmup epochs for varying sampling times is 5 for all datasets.

## B.1 REGULARITY COEFFICIENTS

In sparse operations, regularity coefficients $\lambda_1$ and $\lambda_2$ are often closely related to performance contributions. We set $\lambda_2 = 0$ because $\ell_1$ regularization already induces sparsity directly. The addition of $\lambda_2$ retains more small weights, which may interfere with sparsity. As for adjusting $\lambda_1$, we report the performance of different $\lambda_1$ for the optimal architecture on NTU RGB+D.

Table 7: The contribution of different $\lambda_1$ to sparse operations on NTU RGB+D.

| $\lambda_1$ | S-Attn | S-GLU | S-Concat | S-SELayer |
|---|---|---|---|---|
| 0.1 | 0.04 | 0.25 | 0.08 | 0.33 |
| 0.2 | 0.06 | 0.24 | 0.10 | 0.29 |
| 0.5 | 0.05 | 0.17 | 0.11 | 0.18 |
| 1.0 | 0.03 | 0.12 | 0.06 | 0.15 |
| 1.5 | 0.03 | 0.05 | 0.03 | 0.10 |

In Table 7, we report the contribution of sparse operations. As $\lambda_1$ increases, the contribution of different sparse operations are generally declines, but Sparse-GLU and Sparse-SELayer consistently outperform other sparse operations. Furthermore, sparse operations contribute less than normal operations when $\lambda_1$ is too large. As Fig. 4 in the main text shown, Sparse-GLU and Sparse-SELayer contribute more to the architectures on NTU RGB+D. Therefore, $\lambda_1 = 0.1$ clearly reflects the advantages of sparse operations.

## B.2 SAMPLING TIMES

Table 8: Effect of the number of initial samples on NTU-RGB+D.

| Normal samples $m_n$ | Sparse samples $m_s$ | Accuracy | Search cost (GPU Days) |
|---|---|---|---|
| 10 | 20 | 91.28% | 1.95 |
| 20 | 20 | 92.00% | 2.24 |
| 20 | 40 | 92.67% | 2.50 |
| 20 | 60 | 92.76% | 2.69 |
| 30 | 60 | 92.81% | 3.08 |

Table 8 presents the accuracy and search cost with different number of samples on NTU-RGB+D. We initialize $m_n = 20$ and $m_s = 60$ to balance accuracy and search cost.

## B.3 CELLS AND STEPS

As the key hyperparameters, cells and steps have a great impact on the performance of SM-ShapNAS. For each of the three datasets, we used 2 fusion cells. This is because the proposed method needs to handle low-level feature interactions while capturing global semantic associations across modalities. As the number of cells increases, the performance is not significantly improved, but the search cost is greatly increased. In MM-IMDB, we set 1 step per cell because the multilabel classification task requires less cross-modal interaction between Image and Text, and it is sufficient to complete the basic feature fusion. In NTU RGB+D, Video and Pose sequences are temporally related, and the combined semantics of local action segments (e.g., "wave" + "walk") need to be modeled. 2 steps per cell allows for hierarchical feature fusion. In EgoGesture, RGB and Depth modalities are highly correlated, and require fine-grained alignment. SM-ShapNAS needs to distinguish small differences between categories, and 3 steps per cell is employed to support multistage fusion. Meanwhile, the sparse operations and grouped Shapley value can ensure the explainability of the architectures and reduce the risk of overfitting. Additionally, the number of cells and steps is the same as in BM-NAS, thus better reflecting the advantages of SM-ShapNAS.

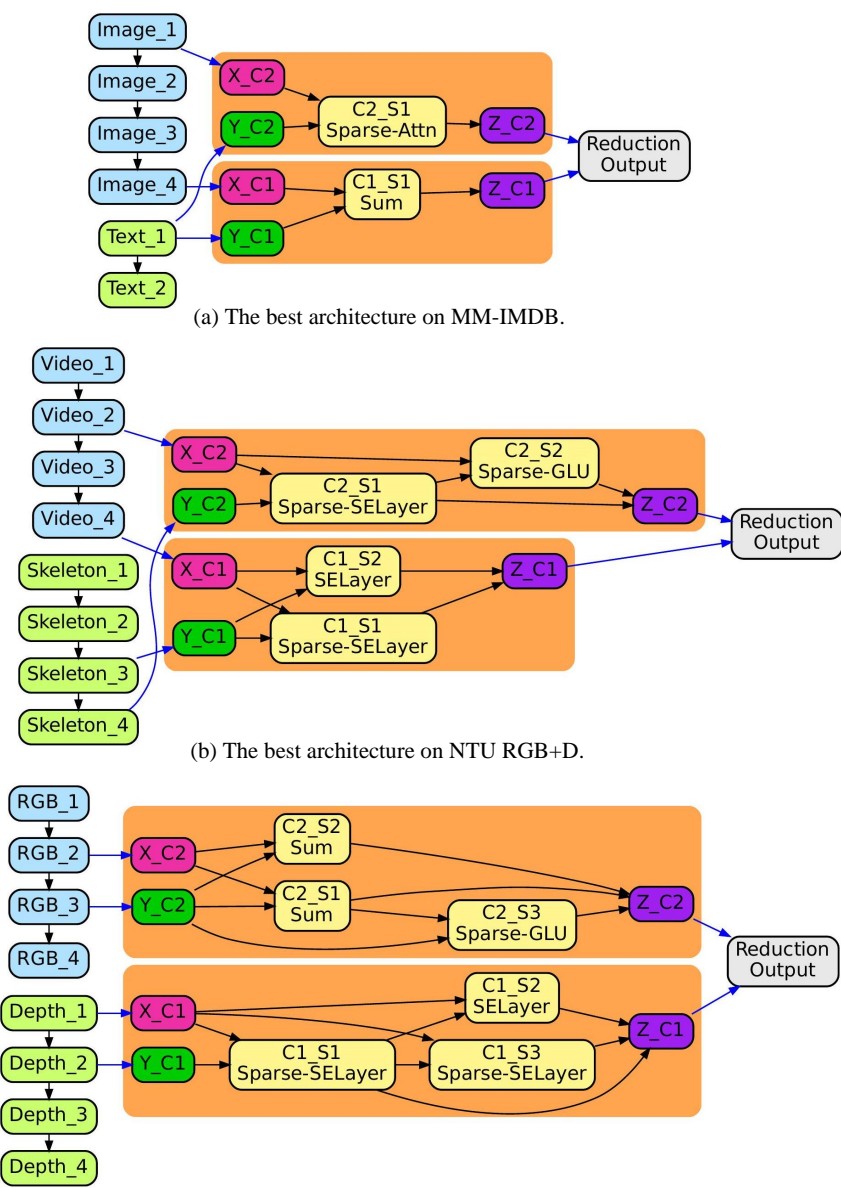

(a) The best architecture on MM-IMDB.

(b) The best architecture on NTU RGB+D.

(c) The best architecture on EgoGesture.

Figure 5: Best architecture of SM-ShapNAS on three multimodal datasets.

## C EXTENDED ANALYSIS

In Table 9, we use better performing backbone for RGB data and multimodal data. The performance is not significant only when the accuracy of one modality is much higher than the other. In fact, the proposed method focuses on multimodal fusion process whose inputs are features generated by unimodal training, therefore replacing the backbone does not lead to a decrease in search efficiency.

Fig. 5 shows the optimal architectures on the three multimodal datasets. The sparse operations are always selected when steps per cell is not less than 2, exhibiting a significantly higher proportion in the NTU RGB+D dataset. This indicates that sparse operations are more suitable for large datasets and complex modal fusion.

Table 9: Performance changes with different backbones on EgoGesture.

| Backbone | RGB (%) | Multimodal (%) |
|---|---|---|
| ResNeXt-101 + ResNeXt-101 | 93.75 | 96.46 |
| Swin-B + Swin-B | 95.38 | 97.54 |
| Swin-B + ResNeXt-101 | 95.38 | 97.13 |

We also test the performance with different probabilities $p_d$ of dropping random modality in Table 10. SM-ShapNAS still performs well when $p_d = 0.1$. As $p_d$ increases, the accuracy decreases rapidly. The performance of the proposed method approximates that of using the Depth modality alone.

Table 10: Results with different $p_d$ on EgoGesture.

| $p_d$ | Acc (%) |
|---|---|
| 0.0 | 96.46 |
| 0.1 | 95.83 |
| 0.2 | 95.33 |
| 0.3 | 94.05 |

Table 11: Modality contribution under varying severity levels of Gaussian noise.

| Noise | Modality | MM-IMDB(%) | NTU RGB+D(%) | EgoGesture(%) |
|---|---|---|---|---|
| Pure | Image/Video/RGB | 60.52 (1.71↑) | 53.89 (1.59↑) | 70.99 (0.89↑) |
| Level 1 | Image/Video/RGB | 57.67 (14.63↑) | 51.64 (2.96↑) | 66.20 (5.95↑) |
| Level 2 | Image/Video/RGB | 25.83 (23.91↑) | 43.00 (7.71↑) | 51.62 (12.90↑) |
| Level 3 | Image/Video/RGB | 14.94 (29.49↑) | 37.50 (11.36↑) | 42.91 (17.34↑) |

The contribution of each modality is shown in Table 11. ↑ represents the proposed method achieves higher results than baseline on three multimodal datasets. As the noise level increases, the utilization of the chosen modality in baseline exhibits a sharp decline, especially in MM-IMDB which does not use pretrained backbone. SM-ShapNAS maintains a stable utilization of the image modality under the same noise conditions. This trend provides evidence that the proposed method has superior capability to extract effective features from degraded inputs.

## D   THE USE OF LARGE LANGUAGE MODELS (LLMS)

This paper did not use LLMs in paper writing.

