# OpenReview forum: "SM-ShapNAS: Shapley-Enhanced Multimodal Neural Architecture Search via Sparse Modeling"
_ICLR.cc/2026/Conference — ICLR 2026 Conference Withdrawn Submission_

### Official Review · Reviewer_HLmT · 2025-10-20

**Soundness:** 2
**Presentation:** 2
**Contribution:** 2
**Rating:** 4
**Confidence:** 4

**Summary:**

This paper proposes a multimodal neural architecture search method for automating the design of appropriate and explainable multimodal DNNs. Specifically, the proposed method incorporates sparse attention and sparse convolutional operations within a predefined search space, and uses the Shapley value approximated by group policy to evaluate the true contribution of each operation in the fusion cells.

**Strengths:**

Using shaplely value is a great choice to make NAS more explainable.

**Weaknesses:**

1. The motivation of sparse modeling is not clearly clarified. Why the sparse modeling is a necessity? Please clarify its motivation.

2. P in Algorithm is not defined. Besides, the input in Algorithm is missing.

3. The explanation for the physical meaning of the first part of Eq. (8) is missing. What is the definition of $P(group = kg)$?

4. How about the Monte Carlo sampling error for Eq. (9)? Authors should experimentally evaluate the stability of the Monte Carlo sampling and the magnititude of such sample error to ensure the reliability of the sampling.

5. Authors do not explain clearly how to get Eqs. (10)--(12). Please clarify clearly.

6. The experiments are limited to two modalities per task. Can the proposed method generalize to  tasks with three or more modalities, such as video + audio + text ?

7. The experiments focus heavily on comparing with other NAS methods but lack comparison with strong hand-designed multimodal models or non-NAS explainable models. This makes it difficult to assess the absolute advantage of the proposed framework.

**Questions:**

Please refer to weaknesses, especially the intuition of sparse modeling and approximation details.
If all my problems are addressed, I will increase my rating.

---

### Official Review · Reviewer_U852 · 2025-10-31

**Soundness:** 1
**Presentation:** 1
**Contribution:** 2
**Rating:** 2
**Confidence:** 4

**Summary:**

The paper introduces SM-ShapNAS to address the challenge of designing explainable and efficient deep neural networks. The method combines the concept of sparse modeling and Shapley values. SM-ShapNAS showed higher performance compared to other NAS methods. While the paper proposes a novel approach to the problem, the low readability makes it hard to grasp the details and the key contributions.

**Strengths:**

- SM-ShapNAS presents a novel approach of integrating two powerful explainability techniques, sparse modeling and Shapley values, into a multimodal NAS framework.

**Weaknesses:**

- The readability of the paper is low, and it was hard to follow the details and the key contributions. Many context-specific jargons were used without any definition or clarification.
- When comparing with other baselines (e.g., in table 2), the control for model sizes is lacking.
- The search space is narrow, and its generalizability or practical implication are limited.

**Questions:**

- What is the definition of “cell” or “fusion cell”?
- “Algorithm 1 shows the search process, which follows DARTS to alternatively optimize architecture parameters and model weight” What does DARTS mean here?
- What does group policy mean?
- Why does the complexity of Shapley value computation go down from O(N · 2^N) to O(mN) after grouping? I understand that the estimates could converge faster as the number of players has decreased, but why does the exponential term also completely disappear?
- Can you discuss the specific advantages and niche of this NAS-based approach compared to other popular strategies like training large models and subsequent pruning, especially for practical implications?

---

### Official Review · Reviewer_4e7q · 2025-10-31

**Soundness:** 3
**Presentation:** 2
**Contribution:** 3
**Rating:** 6
**Confidence:** 2

**Summary:**

This paper proposes SM-ShapNAS, a novel explainable multimodal neural architecture search (NAS) framework that integrates sparse modeling and grouped Shapley value estimation.

Its main contributions are:

(1) a search space incorporating both normal and sparse operations (e.g., sparse attention and sparse convolution) to enhance feature extraction and robustness;

(2) an efficient evaluation mechanism that approximates the Shapley value via group policy and Monte Carlo sampling to fairly quantify each operation’s true contribution;

 (3) an end-to-end differentiable NAS framework that updates architecture parameters based on estimated Shapley values rather than gradient magnitudes;

(4) extensive experiments on three multimodal benchmarks (MM-IMDB, NTU RGB+D, EgoGesture) demonstrating state-of-the-art performance, particularly under noisy conditions, along with comprehensive ablation studies validating the effectiveness of both sparse modeling and the grouped Shapley strategy.

**Strengths:**

The work demonstrates strong originality through a creative synthesis of two theoretically grounded concepts—sparse modeling and Shapley value attribution—within the context of multimodal NAS, a domain where explainability has been largely overlooked. While sparse operations and Shapley-based evaluation have appeared separately in prior literature (e.g., Shapley-NAS for unimodal tasks, sparse coding for interpretability), their joint integration into a differentiable multimodal NAS framework is novel.

The paper is well-structured and clearly written. The narrative flows logically from problem motivation to technical design (search space, Shapley evaluation) to validation (experiments, ablations, visualizations).

**Weaknesses:**

Noise is only applied to image modalities, but real-world multimodal systems often suffer from asynchronous or modality-specific corruption (e.g., missing text, noisy depth, occluded poses). The claim of “robustness in noisy environments” would be stronger if add experiments with cross-modality noise or modality dropout, aligning with recent robust multimodal benchmarks.

**Questions:**

While SM-ShapNAS is a promising step toward explainable multimodal NAS, its claims of robustness, efficiency, and explainability require deeper validation. Addressing the above points like broader noise scenarios and quantitative explainability metrics would significantly strengthen the contribution and align it more closely with its stated goals.

---

### Official Review · Reviewer_cfuV · 2025-11-03

**Soundness:** 2
**Presentation:** 2
**Contribution:** 2
**Rating:** 2
**Confidence:** 4

**Summary:**

This paper proposes SM-ShapNAS, a multimodal neural architecture search (NAS) framework that aims to improve both efficiency and explainability by combining sparse modeling and Shapley-value–based operation evaluation. Within a predefined search space containing both normal and sparse operations, the method estimates grouped Shapley values (via Monte Carlo sampling and group policy) to assess each operation’s contribution to the overall architecture. Sparse convolution and sparse attention layers are included to enhance noise robustness and model interpretability. Experiments on three multimodal datasets show competitive or slightly improved performance compared to previous NAS baselines such as BM-NAS and DC-NAS.

**Strengths:**

The paper makes a clear attempt to integrate explainability concepts into multimodal NAS. Using Shapley values to quantify operation contributions is conceptually coherent with the motivation of fair attribution, and adopting sparse operations is a reasonable choice for improving robustness under noise. The framework is complete, with explicit pseudocode, mathematical definitions, and ablation studies. The implementation appears reproducible, and experiments span multiple datasets, offering empirical breadth.

**Weaknesses:**

- The paper’s technical novelty is limited and its claimed interpretability remains superficial. The use of Shapley values to evaluate architecture components has already appeared in prior work such as Shapley-NAS (Xiao et al., 2022), and this paper extends it mainly by grouping operations and applying Monte Carlo estimation—both standard approximations that add little methodological depth.

- The “group policy” mechanism is under-explained and lacks formal analysis of its convergence or bias.

- The integration of sparse modeling is mostly additive: the sparse attention and sparse convolution layers follow standard formulations and do not fundamentally change the NAS optimization process. As a result, the contribution lies more in combining existing ideas rather than introducing a new algorithmic insight.

- The theoretical framing of the Shapley evaluation is also weakly justified. The architecture-search process still depends on differentiable relaxation (as in DARTS), but the grouped Shapley update (Eq. 10–12 in the paper) is grafted onto this process without explaining how it interacts with gradient-based optimization or what stability guarantees it provides. The cooperative-game analogy is conceptually appealing but mathematically shallow, as the payoff function (validation accuracy) is treated as a noisy, non-additive measure that invalidates the linear-value assumptions of Shapley theory. Furthermore, the paper does not demonstrate that the derived architectures are actually more explainable—no visualization, attribution trace, or qualitative interpretation is shown to support this claim.

- Empirically, the reported gains over BM-NAS and DC-NAS are marginal (e.g., +0.5–1 % on most datasets), and the claimed noise robustness lacks statistical significance testing. Many experimental settings reuse strong pretrained unimodal backbones, meaning improvements could come from data handling rather than the NAS procedure itself. The discussion sections (especially 4.3 and 4.4) mostly restate numerical comparisons without deeper insight into why grouped Shapley evaluation changes the search dynamics.

**Questions:**

- How does the grouped Shapley update interact with gradient-based optimization in DARTS? Does it replace or augment alpha-updates, and what guarantees of convergence or monotonic improvement exist?

- The cooperative-game analogy assumes additivity of payoff contributions. How is this justified when operation effects are non-linear and interdependent?

- Can the authors provide any interpretability evidence (e.g., feature-importance visualizations or contribution maps) beyond scalar fidelity metrics?

- How sensitive are the results to the number of Monte Carlo samples (mn, ms) and the grouping scheme? Are these hyperparameters tuned per dataset?

- Does the proposed framework generalize beyond two-modality fusion (e.g., tri-modal or cross-temporal settings), and how would computational cost scale?

**Details Of Ethics Concerns:**

None.

---

### Note · Authors · 2025-11-13

I have read and agree with the venue's withdrawal policy on behalf of myself and my co-authors.